# Mitochondrial Protein Density, Biomass, and Bioenergetics as Predictors for the Efficacy of Glioma Treatments

**DOI:** 10.3390/ijms25137038

**Published:** 2024-06-27

**Authors:** Gulnaz Sharapova, Sirina Sabirova, Marina Gomzikova, Anna Brichkina, Nick A Barlev, Natalia V Kalacheva, Albert Rizvanov, Nikita Markov, Hans-Uwe Simon

**Affiliations:** 1Laboratory of Molecular Immunology, Institute of Fundamental Medicine and Biology, Kazan Federal University, 420008 Kazan, Russia; gulnazs291@gmail.com (G.S.); sirinakurbangaleeva@gmail.com (S.S.); marina.gomzikova.gmo@gmail.com (M.G.); anna.brichkina@staff.uni-marburg.de (A.B.); nick.a.barlev@gmail.com (N.A.B.); 2OpenLab Gene and Cell Technology, Institute of Fundamental Medicine and Biology, Kazan Federal University, 420008 Kazan, Russia; nvkalacheva@ya.ru (N.V.K.); rizvanov@gmail.com (A.R.); 3Laboratory of Intercellular Communication, Kazan Federal University, 420111 Kazan, Russia; 4Institute of Systems Immunology, Center for Tumor Biology and Immunology, Philipps University of Marburg, 35043 Marburg, Germany; 5Gene Expression Program, Institute of Cytology RAS, 194064 Saint-Petersburg, Russia; 6Department of Biomedical Sciences, School of Medicine, Nazarbayev University, Astana 010000, Kazakhstan; 7Division of Medical and Biological Sciences, Tatarstan Academy of Sciences, 420111 Kazan, Russia; 8I.K. Akhunbaev Kyrgyz State Medical Academy, Bishkek 720020, Kyrgyzstan; 9Institute of Pharmacology, University of Bern, 3010 Bern, Switzerland; 10Institute of Biochemistry, Brandenburg Medical School, 16816 Neuruppin, Germany

**Keywords:** mitochondria, OXPHOS, bioenergetics, metabolism, glioma, glioblastoma, metabolic reprogramming, mitochondria-targeting, drug response, cancer treatment

## Abstract

The metabolism of glioma cells exhibits significant heterogeneity and is partially responsible for treatment outcomes. Given this variability, we hypothesized that the effectiveness of treatments targeting various metabolic pathways depends on the bioenergetic profiles and mitochondrial status of glioma cells. To this end, we analyzed mitochondrial biomass, mitochondrial protein density, oxidative phosphorylation (OXPHOS), and glycolysis in a panel of eight glioma cell lines. Our findings revealed considerable variability: mitochondrial biomass varied by up to 3.2-fold, the density of mitochondrial proteins by up to 2.1-fold, and OXPHOS levels by up to 7.3-fold across the cell lines. Subsequently, we stratified glioma cell lines based on their mitochondrial status, OXPHOS, and bioenergetic fitness. Following this stratification, we utilized 16 compounds targeting key bioenergetic, mitochondrial, and related pathways to analyze the associations between induced changes in cell numbers, proliferation, and apoptosis with respect to their steady-state mitochondrial and bioenergetic metrics. Remarkably, a significant fraction of the treatments showed strong correlations with mitochondrial biomass and the density of mitochondrial proteins, suggesting that mitochondrial status may reflect glioma cell sensitivity to specific treatments. Overall, our results indicate that mitochondrial status and bioenergetics are linked to the efficacy of treatments targeting metabolic pathways in glioma.

## 1. Introduction

One of the fundamental characteristics of cancer cells is metabolic reprogramming, which facilitates their rapid growth and proliferation [1,2,3,4,5]. Conversely, metabolic reprogramming increases the reliance of cancer cells on particular pathways and metabolic axes, thereby hypothetically increasing their susceptibility to treatments targeting these processes. Thus, metabolic adaptations in cancer might serve as a double-edged sword. While they facilitate adaptation, growth, proliferation, and dissemination of cancer cells, these distinct metabolic features also set cancer cells apart, enabling targeted treatment with reduced collateral damage to healthy tissues. 

This reprogramming in cancer cells is tightly linked to mitochondria owing to the fact that the tricarboxylic acid (TCA) cycle and adjacent pathways are heavily integrated into cellular metabolism. Furthermore, mitochondria play a pivotal role not only in cellular bioenergetics but also in a range of other processes crucial for cell maintenance, including redox balance, calcium signaling, lipid and nucleotide synthesis, as well as epigenetic modifications. Alterations and intensifications of these processes are common in cancer cells [6,7,8]. 

Mitochondrial protein composition can vary markedly between different cell subtypes, and even between the resting and activated states of immune cells, indicating a high potential for functional specialization within these organelles [9,10,11,12,13]. Moreover, an increasing amount of evidence indicates that mitochondria are not static organelles, rather they can change their morphology, number, and function and adapt depending on environmental conditions, which can be beneficial for cancer cells under different stresses like nutrient deprivation or hypoxia [8,14,15]. Because of these multifaceted roles, targeting mitochondrial metabolism and homeostasis is considered a promising strategy for cancer therapy. Importantly, while the Warburg effect is still considered a feature of many cancer cells, which prevents them from fully taking advantage of mitochondrial respiration, recent studies clarified that not all cancer cells exhibit this effect [16,17,18]. Moreover, some cancers, including certain types of glioma, rely significantly on oxidative phosphorylation (OXPHOS) for energy production.

Gliomas are tumors that originate from glial cells, the supportive cells of the nervous tissue responsible for maintaining homeostasis, immune responses, and other functions within the central nervous system. Glial cells encompass several different subtypes, including astrocytes, oligodendrocytes, microglia, and ependymal cells. However, only oligodendrocytes, astrocytes, and ependymal cells are known to give rise to gliomas, which are classified as oligodendrogliomas, astrocytomas, and ependymomas, respectively. Glioblastoma (GBM), the most aggressive type of glioma, predominantly arises from astrocytes. The vast heterogeneity of these tumors is influenced by both their cellular origin and their mutational landscapes [19,20,21]. Numerous efforts have been made to develop a nomenclature and classification system based on histology, mutation profiles, chromosomal aberrations, transcriptomics, and other omics approaches [22,23]. The most recent and widely used classification is the 2021 WHO CNS5 [24].

Glioma cells exhibit significant metabolic adaptations and often harbor mutations in key genes regulating metabolic fluxes, such as EGFR, PTEN, and PI3K [25,26]. These mutations frequently lead to the upregulation of MYC transcription factor activity and the AKT-mTOR signaling pathway, thereby enhancing anabolic processes within the cells. Notably, the “signature” oncogenic mutations in gliomas occur in the IDH1/IDH2 genes, which are directly involved in the mitochondrial reactions of the tricarboxylic acid (TCA) cycle [25]. GBM cells are capable of utilizing both glycolysis and oxidative phosphorylation OXPHOS in the mitochondria during aggressive tumor growth [27,28,29]. Importantly, 13C nuclear magnetic resonance has revealed that the majority of acetyl-CoA in patients with GBM and mouse GBM xenografts is not derived from glucose, suggesting that additional carbon sources contribute to GBM bioenergetics. Recent studies have indicated that acetate might be one of these supplementary fuels, being metabolized into acetyl-CoA in these cells [30].

Considering the significant metabolic and bioenergetic plasticity of glioma cells, we hypothesized that the efficacy of therapies targeting these cells would depend on their metabolic dependencies. Our rationale was straightforward: we assumed that cells displaying high levels of OXPHOS would be more susceptible to therapies targeting mitochondria; conversely, cells with increased glycolytic flux would be more susceptible to inhibitors of glycolysis. Furthermore, we aimed to investigate whether there is a connection between mitochondrial status/bioenergetics and the efficacy of conventional glioma treatments such as temozolomide and carmustine.

To test these hypotheses, we employed a panel of eight glioma cell lines. Initially, we characterized the mitochondrial biomass, density of mitochondrial proteins, rates of OXPHOS, and glycolysis in these cell lines, subsequently categorizing them into distinct mitochondrial and bioenergetic subtypes. After stratifying the cell lines, we evaluated the effects of 16 compounds that interfere with various metabolic and homeostatic pathways. The results from these experiments clarified relationships and tendencies between mitochondria and metabolic statuses of glioma cells and their sensitivity to treatments targeting metabolic pathways. 

## 2. Results

### 2.1. Mitochondrial Biomass and Density of Mitochondrial Proteins Are Linked with Proliferation Rate of Glioma Cells 

To explore mitochondrial variability in diffuse glioma, we assembled a panel of eight commonly used glioma cell lines (LN319, LN428, T98G, A172, LN18, U251, LN229, D247MG). These cell lines originate from three primary types of gliomas, namely, glioblastoma, astrocytoma, and gliosarcoma, thus representing the main types of diffuse glioma tumors, excluding oligodendrogliomas and mixed oligoastrocytomas (Table 1). 

The selected cell lines displayed substantial variability in key oncogenic mutations in genes such as TP53, PTEN, NF1, EGFR, IDH1/2, and PIK3R1. For instance, cell lines LN319, LN428, and U251 harbored mutations in at least three of these genes, whereas cell lines A172, LN18, and D247MG either lacked mutations in these genes or had mutations in only one of them, indicating diverse mutational compositions and adaptations driving these cancer subtypes.

Considering major differences in the origin and mutational burden of the selected cell lines, we assessed the proliferation rate of the examined cells in order to have a reference understanding of their “aggressiveness”, considering the fact that the biological success of cancer is at least partially attributed to its rapid proliferation capabilities (Figure 1A). Surprisingly, the proliferation rates varied significantly among the examined cells, with up to a fourfold difference between A172 and LN18. Notably, the A172 and D247MG cell lines, which were characterized by a slower proliferation rate, were the only ones possessing wild-type p53, aligning with the current understanding that mutant versions of p53 tend to enhance cancer cell proliferation rates [31,32].

Next, to explore potential relationships between mitochondrial abundance and proliferation in glioma cells, we employed MitoTracker Green and MitoTracker Deep Red dyes, which specifically stain mitochondria (Figure 1B). We used each of these dyes at two different concentrations to minimize potential bias from unspecific staining of other organelles. Intriguingly, the pooled results from both dyes showed very similar outcomes, indicating that individual glioma cell lines generally have a comparable and stable abundance of mitochondria. In contrast, we detected substantial inter-cell line variability in mitochondrial content with a signal ranging from as low as 48% to as high as 155% relative to the average level.

Mitochondria are known to vary significantly in their protein composition [10,11]. Additionally, cells can accumulate mitochondria with varying densities of mitochondrial proteins, which may exhibit increased or decreased functional capabilities [33,34,35]. In some cases, cells may also accumulate dysfunctional mitochondria due to impairments in mitophagy [36,37]. Considering these factors, we decided to investigate the density of mitochondrial proteins in the examined glioma cell lines. We assessed the abundance of five proteins that compose the electron transport chain (ETC) complexes I, II, III, IV, and V, along with TOMM20, which is involved in mitochondrial protein import (Figure 1C, Appendix A). 

The Western blot analysis revealed that cell lines such as A172, LN229, and particularly, D247MG, displayed a decreased abundance of mitochondrial proteins. Notably, while D247MG cells contained the highest quantity of mitochondria, these mitochondria exhibited a reduced density of some essential ETC proteins, suggesting the presence of mitochondrial dysfunction in these cells.

Intrigued by the observed differences in mitochondrial biomass and the density of mitochondrial proteins, we conducted a correlation analysis between these parameters and the proliferation rates of glioma cell lines (Figure 1D,E). Strikingly, both comparisons revealed strong correlations, albeit in contrasting directions. While mitochondrial biomass was negatively associated with proliferation rate, the density of mitochondrial proteins showed a positive correlation with this parameter. This data suggests that the most effective strategy for enhancing cellular proliferation might involve decreasing the absolute number of mitochondria while increasing the abundance of proteins within these organelles.

Finally, we assessed the activation levels of AKT and AMPK, proteins that control the activity of major anabolic and catabolic pathways converging on the same TSC1/2 complex and significantly influencing proliferation levels (Figure 1F), [38,39]. Activation of these proteins is known to depend on cellular energy and growth factors and is linked with mitochondrial functionality [40,41,42]. Our objective was to determine whether there is a connection between mitochondrial biomass/density and the activation of this regulatory axis controlling cellular anabolism/catabolism and proliferation in cells. However, the activation patterns of these proteins did not correlate with the cell proliferation rates. For example, despite having the slowest proliferation rate and absence of oncogenic mutations in the PI3K-AKT pathway, D247MG cells exhibited high AKT signaling and moderate AMPK activation, displaying a major discrepancy between the activity of these signaling pathways and expected and observed proliferative behavior.

### 2.2. Glioma Cell Lines Display Major Variability in OXPHOS Levels 

Intrigued by the variations in the density of mitochondrial proteins and biomass in glioma cells and their association with proliferation rates, we sought to determine whether these changes impact the rates of mitochondrial respiration and OXPHOS. Therefore, we evaluated parameters related to oxygen consumption and mitochondrial respiration, following sequential injections of oligomycin, FCCP, rotenone + antimycin A, and 2-deoxy-D-glucose (2-DG) (Figure 2A–D). Our bioenergetic flux analysis revealed that D247MG cells exhibited the lowest levels of mitochondrial respiration parameters, suggesting the presence of mitochondrial dysfunction in this cell line. Additionally, the glioma cell lines displayed highly variable levels of OXPHOS, indicating strong potential to use this parameter to stratify glioma cells in both research and clinical studies. 

Furthermore, the distribution of OXPHOS levels demonstrated that both glioblastoma and astrocytoma cell lines can vary significantly in terms of their mitochondrial respiration, displaying both increased and decreased levels relative to the average.

Complimentary to mitochondrial respiration profiling, we also assessed bioenergetic parameters related to glycolysis in glioma cells (Figure 2E–G). Interestingly, and contrary to our expectations, glioma cells with reduced OXPHOS did not significantly upregulate their glycolysis as a compensatory mechanism. Moreover, overall glycolysis rates were more consistent across the examined cells compared to OXPHOS, suggesting that glycolysis rates are less variable in glioma cells.

We summarized the results of the bioenergetic analysis using OCR/ECAR ratios to illustrate the metabolic dependencies of the examined glioma cells (Figure 2H). The results showed that almost all the examined cells, except D247MG, which displayed properties of mitochondrial dysfunction, displayed similar OCR/ECAR ratios, indicating that glioma cells generally favor similar balances between OXPHOS and glycolysis. However, despite these similar ratios, the intensities of these bioenergetic processes, especially mitochondrial respiration, can vary significantly among the cell lines.

To represent the combined intensity of these processes, we calculated the sums of mitochondrial respiration and glycolysis, using these parameters as indicators of the overall “bioenergetic fitness” of the examined glioma cells (Figure 2I). This analysis revealed substantial variability in the bioenergetic capacities of the glioma cells, primarily driven by differences in mitochondrial respiration. Additionally, we plotted corresponding OCR and ECAR values to examine the bioenergetic separation between the cell lines (Figure 2J). This graph clearly illustrated a nearly linear relationship between OXPHOS and glycolysis in glioma cells, where an increase in OXPHOS is almost invariably accompanied by an increase in glycolysis. 

Collectively, this analysis highlights the significant variation in the bioenergetic activity of glioma cells and suggests that one of the most rational ways to bioenergetically categorize them is by stratifying the cells based on the sum of mitochondrial respiration and glycolysis, which we titled “bioenergetic fitness” in this study.

### 2.3. Susceptibility of Glioma Cell Lines to Mitochondria-Targeting Is Associated with Bioenergetics and Mitochondrial Parameters 

Considering the substantial differences in mitochondrial status and bioenergetics of glioma cells, we hypothesized that the efficiency of therapeutic agents targeting various mitochondrial, bioenergetic, and metabolic pathways depends on the mitochondrial status and intensity of bioenergetic processes. Specifically, we assumed that cells with either high or low mitochondrial respiration, density of mitochondrial proteins, mitochondrial biomass, glycolysis, or overall bioenergetic fitness would respond differently to inhibition of these and adjacent pathways. In order to test this, we treated all examined glioma cells with 16 compounds/drugs targeting various cellular processes related to bioenergetics, metabolism, and mitochondria (Appendix A). The selected concentrations of these compounds were chosen primarily for the following reasons: they are within the range commonly used in other studies and we performed pilot experiments with different concentrations where the results indicated that these concentrations most effectively separate the cells in terms of their drug responses.

To assess the drug response to treatments, we chose a time point of 72 h post-treatment. This relatively late time point was selected because our primary interest was in understanding the long-term tendencies and consequences rather than rapid initial responses. To evaluate the cellular drug responses, we utilized three assays measuring the number of cells, apoptosis levels, and proliferation rates. Given that changes in cell number are predominantly influenced by two main factors such as cell proliferation and cell death, we selected these assays as they can evaluate the rates of these processes and thus identify the primary reasons for changes in cell number. The raw values and data distribution corresponding to these assays for all 16 treatments across all examined cell lines are presented in Appendix A. In the main figures, all values were normalized as percentages or fold changes relative to untreated cells. To explore the associations between drug-induced changes in cell number, apoptosis, and proliferation with baseline mitochondrial and bioenergetic characteristics, we calculated correlation matrices that evaluated the relationships between these parameters (see Appendix A).

Initially, to investigate the responses of cells with different levels of OXPHOS to the ETC inhibition, we utilized treatments such as rotenone, antimycin A, and oligomycin, which target complexes I, III, and V of the ETC, respectively (Figure 3A–C). 

All three compounds, particularly antimycin A, demonstrated noticeable toxicity at nanomolar concentrations, indicating that glioma cells are highly sensitive to ETC targeting. Correlation analysis revealed that the effectiveness of rotenone and antimycin A treatments (measured by their ability to reduce cell numbers through the inhibition of proliferation and induction of apoptosis) was positively associated with the density of mitochondrial proteins, suggesting that this parameter can predict the response to these treatments. Interestingly, despite the direct involvement of complex V in ATP production, the inhibition of this complex by oligomycin did not show any significant associations with mitochondrial features or overall bioenergetics. Collectively, these results suggest that glioma cells with high levels of mitochondrial respiration are not necessarily more sensitive to ETC and OXPHOS inhibition than those with lower respiration levels.

Furthermore, the inhibition of mitochondrial fusion/fission dynamics [43,44] induced by mdivi-1, a crucial process for proper mitochondrial function, resulted in a major decrease in glioma cell numbers (Figure 3D). Notably, compared to most other treatments, the cytotoxicity of mdivi-1 was mainly driven by the induction of apoptosis. The efficacy of mdivi-1 treatment was clearly associated with decreased mitochondrial biomass and increased density of mitochondrial proteins. This suggests that mdivi-1 is particularly effective against cells that have a low number of mitochondria, which display high functional capacity.

To maintain sufficient mitochondrial membrane potential (MMP), cells need to sustain robust activity of the TCA cycle, which they fuel using various metabolic pathways. Three key pathways that are especially critical are mitochondrial pyruvate import, fatty acid oxidation (FAO), and glutaminolysis [45,46]. Moreover, it is increasingly recognized that cancer cells can develop dependencies on specific mitochondrial fuels, relying more heavily on certain sources than others [47]. For that reason, we aimed to evaluate mitochondrial dependencies in glioma cells by inhibiting these mitochondrial fluxes. We used etomoxir, UK5099, and BPTES to inhibit FAO, mitochondrial pyruvate import, and glutaminolysis, respectively (Figure 3E–G). Interestingly, while glutaminolysis inhibition induced a reduction in cell numbers by more than 30% in some glioma cell lines, neither FAO nor pyruvate import inhibition achieved a more than 20% reduction in examined cell lines. This suggests that glioma cells can largely compensate for these effects and that inhibition of a single mitochondrial fuel does not fully bottleneck their metabolism.

Furthermore, correlation analysis indicated that inhibition of mitochondrial fuels displays a more profound effect on cells with high mitochondrial biomass but reduced density of mitochondrial proteins. This suggests that cells with high mitochondrial biomass but reduced protein density may be less adaptable to shifts in metabolic pathways caused by the inhibition of key mitochondrial enzymes, and thus, may exhibit a more significant response to such treatments.

In comparison to inhibitors of mitochondrial fuels, 2-DG-induced inhibition of glycolysis elicited a substantial drug response in the majority of glioma cell lines, once again highlighting the critical importance of this pathway for cancer cells (Figure 3H). However, the efficacy of reducing cell numbers varied significantly, ranging from 21% in the case of the LN22 cell line to 88% in the case of the D247MG cell line. The efficacy of this treatment was strongly positively associated with mitochondrial biomass and negatively associated with mitochondrial respiration and the density of mitochondrial proteins. This data suggests that 2-DG is most effective against cancer cells with dysfunctional mitochondria by targeting the primary biological pathway on which they rely.

Interestingly, the D247MG cell line was also the most susceptible to the inhibition of any mitochondrial fuels. This cell line exhibited the lowest level of bioenergetic fitness, suggesting that inhibition of any mitochondrial fuels is particularly critical for this glioma phenotype, highlighting its vulnerability to disruptions in metabolic pathways.

### 2.4. Glioma Cells Are Sensitive to Impairment of Acidification Regulation. The Efficacy of Chemotherapy Treatments Is Associated with the Density of Mitochondrial Proteins 

Next, we investigated the effects of EIPA treatment. EIPA acts as a specific inhibitor of the Na^+^/H^+^ exchangers, which are crucial for regulating intracellular pH by extruding H+ ions in exchange for Na^+^ ions. The inhibition of these systems results in the acidification of the cytoplasm and can lead to cell death. Given that the main sources of H^+^ ions and cellular acidification are glycolysis and mitochondrial respiration, we initially assumed that cells with high levels of these bioenergetic processes would be particularly vulnerable to EIPA treatments. Surprisingly, contrary to our initial expectations, EIPA showed increased efficacy against cells with low mitochondrial respiration or bioenergetic fitness (Figure 4A). One possible reason for this response pattern is that cells with reduced bioenergetic fitness do not possess spare energy to counteract the pH imbalance induced by this treatment.

Additionally, we explored the effects of inhibiting a system involved in cellular acidification, specifically focusing on organelles such as endosomes and lysosomes, which require acidification for proper function. This acidification is maintained by the continuous activity of vacuolar-type H^+^-ATPases. We hypothesized that cells with distinct bioenergetic profiles might exhibit varied responses to the inhibition of organelle acidification. To explore this connection, we used concanamycin A, a selective inhibitor of vacuolar-type H^+^-ATPases. Even at a low concentration of 2 nM, the inhibition of vacuolar-type H^+^-ATPases resulted in drastic cytotoxicity (Figure 4B). The treated glioma cells displayed massive levels of apoptosis induction and proliferation inhibition. Among all the drugs tested in this study, concanamycin A showed the highest overall cytotoxic effect at the lowest concentration. Nonetheless, the impact of this compound was not clearly associated with mitochondrial features or overall bioenergetics.

Furthermore, our objective was to explore potential associations between mitochondrial/bioenergetic characteristics and the effectiveness of conventional chemotherapy used in glioma treatment. In order to achieve this, we assessed the drug response induced by carmustine and temozolomide, one of the most commonly used therapeutics in glioma treatment (Figure 4C,D), [48,49]. Strikingly, even at a relatively high concentration of 20 µM, both treatments were almost completely ineffective against the LN18, D247MG, and T98G cell lines, underscoring the urgent need for the development of new treatment options for this type of cancer. Nonetheless, we observed that none of the examined mitochondrial or bioenergetics characteristics were strongly associated with the efficacy of these drugs, suggesting that examined parameters have low predictive potential for typical anti-glioma treatments. 

Next, we aimed to evaluate how clinically approved agents known for their mitochondrial toxicity affect glioma cells with varying dependencies on mitochondrial respiration. These compounds disrupt various homeostatic processes in mitochondria and may display additional interest for repurposing as anti-cancer therapeutics. Specifically, we examined the effects of cyclosporine A, an immunosuppressant known to impair the function of the mitochondrial permeability transition pore; stavudine, an anti-HIV treatment that also inhibits mitochondrial polymerase γ; and dimethyl fumarate, a derivative of the TCA cycle intermediate fumarate, which is capable of succinating various cellular enzymes and is approved for the treatment of psoriasis and multiple sclerosis [50,51,52]. Interestingly, while none of these compounds induced drastic levels of cytotoxicity, cyclosporine A significantly reduced the cell count of T98G and LN18 cells by nearly 50%, suggesting its potential for repurposing against specific types of gliomas (Appendix A). Therefore, it is crucial to identify the commonalities between LN18 and T98G cells that may render them particularly susceptible to cyclosporine A. Furthermore, the effects of cyclosporine A negatively correlated with mitochondrial biomass, suggesting that it could be potentially effective against cancers characterized by low mitochondrial number.

Additionally, we evaluated the effect of AZD3965, a promising molecule known for inducing an overload of lactate, the terminal product of glycolysis (Appendix A), [53]. This compound is currently being assessed in both in vivo studies and clinical trials, showing promising results [54]. However, in our study, AZD3965 neither exerted a strong effect on glioma cells nor showed noticeable correlations with investigated mitochondrial and bioenergetic characteristics.

Finally, we pooled the results of all treatments to determine if any of the examined glioma cell lines displayed increased or decreased sensitivity to all treatments, which could potentially interfere with the interpretation of associations between drug response and cellular metrics (Appendix A). For instance, we hypothesized that due to its low bioenergetic fitness, the D247MG cell line might be more sensitive on average to any treatment. However, the distribution of cell responses, illustrated by changes in cell numbers, proliferation, and apoptosis, showed similar results across the examined cell lines, suggesting that there are relatively similar levels of average treatment susceptibility among them.

### 2.5. Mitochondrial Parameters and Bioenergetics Are Associated with Response of Glioma Cells to Targeting of Metabolic and Homeostatic Pathways

To consolidate the associations of changes in cell number, proliferation, and apoptosis with any basal metric into a single value, we introduced a combined drug response correlation score. This score is a weighted sum of three Pearson correlation coefficients (Pcc) as detailed in Section 4. We then summarized and visualized the results using this score to illustrate associations between the effects induced by the tested treatments and cellular bioenergetics or mitochondrial parameters (Figure 5A,B). The results indicated that in glioma, none of the treatments consistently exhibited a stronger effect against cells that were more glycolytic or more reliant on mitochondrial respiration. This general trend applied to all drugs except for oligomycin and temozolomide, which showed a minor tendency to be more effective against cells with increased OXPHOS and decreased glycolysis rates. In contrast, the efficacy of mdivi-1, stavudine, and BPTES was predominantly associated with the activity of the glycolysis pathway and was independent of OXPHOS rates.

Furthermore, the majority of other treatments showed increased efficacy against cells with either increased or decreased bioenergetic fitness, which represents an increase or decrease in both bioenergetic characteristics. Specifically, EIPA, antimycin A, concanamycin, and etomoxir displayed increased efficacy against cell lines with reduced bioenergetic capacity and potentially more dormant phenotypes. Conversely, AZD3965 was more effective against metabolically more active cells.

We also explored whether the responses of cells to treatments were more strongly associated with mitochondrial biomass or density of mitochondrial proteins rather than mitochondrial respiration in glioma. To our surprise, the efficacy of almost all analyzed treatments and induced drug responses was linked with one or two mitochondrial metrics. The effectiveness of treatments induced by one group of drugs, including etomoxir, 2-DG, UK5099, BPTES, and AZD3965, was negatively associated with the density of mitochondrial proteins and positively with mitochondrial biomass. This suggests that the ratio of these two parameters could relatively accurately predict the sensitivity of glioma cells to these treatments. Conversely, another group of treatments, represented by cyclosporine A, mdivi-1, antimycin A, and rotenone, was positively associated with the density of mitochondrial proteins and negatively with mitochondrial biomass.

Taken together, this analysis indicates that parameters such as mitochondrial biomass or density of mitochondrial proteins may provide more insightful metrics than general bioenergetics for predicting glioma cell responses to treatments targeting mitochondrial, metabolic, and bioenergetic processes.

## 3. Discussion

Glioblastoma is associated with a poor prognosis, with a median survival of only 14 months [55,56]. Despite initially successful treatment involving surgery, radiotherapy, and chemotherapy, the recurrence rate for glioblastoma is as high as 90% [57]. Several key features contribute to the difficulty in effectively treating this cancer. The diffuse nature of glioblastoma growth limits the possibility of complete surgical removal, often leading to tumor recurrence. Additionally, the presence of the blood–brain barrier restricts many potentially effective drugs from reaching the tumor site at effective concentrations. Another significant challenge is the restricted immune response, attributed to the immune-privileged status of the brain. These factors collectively make the treatment of glioblastoma more challenging compared to other tumors [58,59].

Therefore, glioma treatment urgently requires the development of new therapeutic strategies. One promising approach is targeting the metabolism of cancer cells [60,61]. Novel strategies include focusing on various metabolic processes such as FAO, mitochondrial respiration, ETC reactions, OXPHOS, glutaminolysis, metabolism of amino acids, one-carbon metabolism, cholesterol metabolism, the TCA cycle metabolism, and nucleotide metabolism. Many of these pathways are either directly located in mitochondria or closely linked to mitochondrial biochemical pathways, suggesting that in cancer, mitochondria could represent an Achilles’ heel [62]. This is particularly relevant considering that these organelles host part of the apoptotic machinery, and intrinsic apoptosis is driven by mitochondrial depolarization.

In this study, we assessed the efficacy of various metabolic-targeting strategies as monotherapy against a panel of eight glioma cell lines. Some of these strategies, including inhibitors of glycolysis, the ETC, and glutaminolysis, displayed promising efficacy. They exhibited higher cytotoxicity and a greater reduction in cell number at much lower concentrations than the conventional therapeutics assessed in this study. However, other strategies, such as the inhibition of pyruvate import, FAO, and lactate export, showed relatively low cytotoxicity and were only effective against some of the examined glioma cell lines. This suggests that, at least as monotherapies, these strategies might have limited applicability in the treatment of glioma.

In this study, we linked the efficacy of assessed compounds with the bioenergetic profiles and mitochondrial features of the examined cells. Our findings suggest that the efficacy of certain treatments is closely associated with bioenergetics. For example, treatments such as antimycin A and EIPA were more effective against cells characterized by low OXPHOS and glycolysis rates. More surprisingly, we observed numerous strong associations between the efficacy of treatments and mitochondrial biomass and the density of mitochondrial proteins, suggesting that characteristics defining mitochondrial parameters and mitochondrial “health” could be crucial for predicting treatment outcomes.

Among the most notable associations were the positive correlations linking the effectiveness of 2-DG, etomoxir, UK5099, and BPTES treatments with mitochondrial biomass and negative correlations with the density of mitochondrial proteins. Interestingly, all these compounds inhibit mitochondrial fuels and displayed very similar associations in terms of their effectiveness. Additionally, our data suggested that the density of mitochondrial proteins might serve as a proxy for the overall functional capability of mitochondria, extending beyond just OXPHOS.

Mitochondria are organelles that not only host the OXPHOS machinery necessary for energy production but also contain components of the apoptotic machinery and are central to many key metabolic and signaling pathways [63,64,65]. This positions mitochondria at the core of numerous cellular processes, essentially serving as the heart of the cells in some respects. Given their integral role, the status of mitochondria can significantly reflect the overall cellular condition. This central importance is likely why many treatments show a strong correlation in efficacy with the density of mitochondrial proteins and mitochondrial biomass. Such findings underline mitochondrial characteristics as potential valuable indicators for evaluating the effectiveness of various therapeutic interventions. Additionally, some existing methods for measuring the induction of apoptosis and cell death are based on assessing mitochondrial depolarization and status, further highlighting the central role of mitochondria in cellular health and response to treatments [66,67].

Furthermore, our results indicated that cells with the lowest bioenergetic fitness (LN229 in the case of glycolysis and D247MG in the case of OXPHOS rates) did not display increased susceptibility compared to other cells. This suggests that a low intensity of bioenergetic processes does not automatically render cells more susceptible to treatments.

Interestingly, mitochondria exhibit variable characteristics based on age and gender, displaying differences in certain properties between young and elderly individuals, as well as between males and females [68,69]. Correspondingly, glioblastoma manifests differently in males and females and across various age groups, characterized by distinct sets of mutations that are related to age and gender [70]. This observation suggests a potential connection between mitochondrial alterations that are specific to age and gender and the subtypes of glioblastoma.

Our study has clear limitations: we utilized only one concentration of compounds, and it is possible that different concentrations could yield different results. Moreover, we did not address hypoxic condition, which is known to significantly alter both bioenergetics and mitochondrial function, as well as cell responses [71,72]. Additionally, our study lacks in vivo models and ex vivo collected patient material, which limits our ability to fully capture the complexity of living organisms. Future studies should address these aspects to provide a more comprehensive understanding of the therapeutic potential and cellular responses under varied environmental conditions.

Nevertheless, our findings indicate that the effectiveness of metabolic targeting significantly depends on the rate of the metabolic processes that these treatments target. Therefore, for optimal results, both clinical trials and treatment strategies that utilize metabolic targeting approaches should incorporate patient stratification that accounts for the intensity of metabolic processes. Furthermore, our data suggest that at least in glioma cancer, profiling of mitochondria and stratification of cells based on mitochondrial parameters might be potentially even more insightful and valuable than one on bioenergetic characteristics.

## 4. Materials and Methods

### 4.1. Cell Culture

Authenticated glioma cell lines (LN319, LN428, T98G, A172, LN18, U251, LN229, and D247MG) were generously provided by Prof. Michael Weller [73]. These cell lines were cultured in DMEM media (Thermo Fisher Scientific, Waltham, MA, USA; Cat. # 31966047) containing 100 U/mL of penicillin/streptomycin and 10% FBS at 37 °C in the incubator supplied with 5% of CO_2_. Cells were split every 3–4 days with the usage of trypsin-EDTA solution (Thermo Fisher Scientific, Cat. # 25200072). All experiments were conducted simultaneously across all cell lines to ensure consistency and relevance. Regular mycoplasma testing was performed, and only low-passage cells were used to maintain experimental integrity.

### 4.2. Analysis of Cell Proliferation Rate

Glioma cells were stained using 3 µM CellTrace Violet (Thermo Fisher Scientific, Cat. # C34557), in DPBS supplemented with 0.1% FBS for a duration of 10 min. After staining, the cells were rinsed twice with DPBS containing 10% FBS and subsequently cultured in standard growth media. On the following day, reference control cells were harvested by trypsinization, rinsed, and analyzed using a BD FACSLyric™ flow cytometer (BD Biosciences) to measure the basal fluorescence level, quantified at the experiment’s initial time point (D1). The remaining cells were either maintained as control cells or exposed to various agents, followed by a 72 h incubation. After 72 h, cells were trypsinized, washed, and subjected to flow cytometry analysis. The rate of cellular proliferation was assessed by comparing the fluorescence intensity of the reference control sample at D1 with that observed at the conclusion of the experiment at D4.

### 4.3. Western Blotting

Protein samples were extracted by lysing glioma cells in RIPA buffer containing a protease inhibitor cocktail (Sigma-Aldrich, St. Louis, MI, USA; Cat. # P8340) and 1 mM PMSF. Subsequently, 30 μg of proteins were denatured using 100 mM dithiothreitol and LDS Sample Buffer (Abcam, Cambridge, UK; Cat. # ab119196) and heating. The proteins were separated electrophoretically and transferred onto a PVDF membrane (Merck Millipore, St. Louis, MI, USA; Cat. # IPVH00010). The membranes were blocked with 5% non-fat milk in TBST for 1 h and incubated overnight at 4 °C with primary antibodies, including the Total OXPHOS Human WB Antibody Cocktail (Abcam, Cat. # ab110411), TOMM20 (Abcam, Cat. # ab186735), beta-actin (GenScript, Piscataway, NJ, USA; Cat. # A00730), Ser473-phospho-Akt (Cell Signaling Technology, Danvers, MA, USA; Cat. # 9271), Thr308-phospho-Akt (Cell Signaling Technology Cat. # 9275), Akt (Cell Signaling Technology Cat. # 9272), Thr172-phospho-AMPKα (Cell Signaling Technology Cat. # 2535), and AMPKα (Cell signaling technology Cat. # 2532). Following this, membranes were washed in TBST and treated with HRP-conjugated secondary antibodies (GE Healthcare, Chicago, IL, USA). Protein detection was achieved by inducing chemiluminescence with Clarity Western ECL Substrate (Bio-Rad, Hercules, CA, USA; Cat. # 1705061) and visualized using the ChemiDoc XRS+ system (Bio-Rad)

### 4.4. Quantification of Mitochondrial Biomass

To reduce bias associated with the non-specific accumulation of mitochondrial dyes in organelles other than mitochondria at different concentrations, we employed two distinct mitochondrial stains, each at two different concentrations. In brief, glioma cells were trypsinized, washed, and resuspended in complete media containing either 200 or 1000 nM of MitoTracker™ Green FM (Thermo Fisher Scientific, Cat. # M7514) or 20 or 100 nM of MitoTracker™ Deep Red FM (Thermo Fisher Scientific, Cat. # M22426). The cells were incubated for 30 min at 37 °C, then analyzed using a BD FACSLyric™ flow cytometer (BD Biosciences). For comparative analysis, the acquired data were normalized to the mean and then combined.

### 4.5. Analysis of Cellular Bioenergetic Profiles Using Seahorse XFe96

To ensure minimal interference from external factors, the metabolic profiles of all glioma cell lines were simultaneously measured using the Seahorse XFe96 analyzer. Due to differences in cell size, the number of cells seeded in the XFe96 cell culture microplate varied between 40,000 and 70,000. Results were normalized based on the initial seeding coefficients to account for differences in cell numbers. To reduce variability caused by differing rates of proliferation among the cell lines, cells were seeded on poly-L-lysine coated plates, which promote rapid adhesion and allow for immediate execution of the assay. The assay was performed using XF DMEM Medium pH 7.4 (Agilent, Santa Clara, CA, USA; Cat. # 103575-100), supplemented with 10 mM glucose (Agilent, Cat. # 103577-100), 2 mM L-glutamine (Agilent, Cat. # 103579-100), and 1 mM sodium pyruvate (Agilent, Cat. # 103578-100). Cells were incubated at 37 °C in a non-CO_2_ incubator for 1 h before measurements. The Seahorse assay measured both the oxygen consumption rate (OCR) and extracellular acidification rate (ECAR) under basal conditions and following the sequential injection of metabolic inhibitors at final concentrations of 1.5 μM oligomycin (Sigma-Aldrich Cat. # O4876), 2 μM FCCP (Sigma-Aldrich Cat. # C2920), 1.5 μM rotenone (Sigma-Aldrich Cat. # R8875) with 1.5 μM antimycin A (Sigma-Aldrich Cat. # A8674), and 50 mM 2-deoxy-D-glucose (Sigma-Aldrich Cat. # D6134). The assay enabled the calculation of the following key bioenergetic parameters related to mitochondrial respiration and glycolysis. Mitochondrial respiration: basal mitochondrial respiration (change in OCR from steady state to post-rotenone+antimycin A), ATP-linked respiration (change in OCR from basal to post-oligomycin), and maximal mitochondrial respiration (change in OCR from non-mitochondrial to post-FCCP). Glycolysis: glycolysis (change in ECAR from steady state to post-2-DG) and maximal glycolytic capacity (change in ECAR from non-glycolytic to post-oligomycin).

### 4.6. Quantification of Cell Numbers and Apoptosis Levels

The quantification of cell numbers was performed using the BD FACSLyric™ flow cytometer (BD Biosciences, Franklin Lakes, NJ, USA), which utilizes its fixed-speed-aspiration capability to count cells. Initially, the supernatant containing both floating and dead cells was collected. To inactivate trypsin and prevent potential cell loss, the collected supernatant from each sample was used for cell resuspension. To avoid any loss of cells, the harvested cells were neither washed nor centrifuged. A portion of each sample was then vortexed and immediately measured for a duration of 45 s. The total number of cells in the sample was calculated based on the count obtained from the measured sample fraction, considering only events that corresponded to cells of normal size, with debris excluded.

Apoptotic glioma cells were detected using Annexin V staining. Cells were first trypsinized, then washed and resuspended in 50 µL of Annexin V binding buffer (pH 7.4) containing 10 mM HEPES, 140 mM NaCl, and 2.5 mM CaCl_2_. The cells were then incubated with 1.25 µL of APC-conjugated Annexin V probe (BD Biosciences, Cat. # 550475) for 15 min at room temperature. Following incubation, the cells were analyzed using the BD FACSLyric™ flow cytometer (BD Biosciences) to identify and quantify apoptotic cells.

### 4.7. Statistical Analysis and Calculation of Correlation Matrixes

The visualization of data was carried out using GraphPad Prism 9.5.1 and FlowJo 10.9 software, with the latter also used for the analysis of flow cytometry data. All values are presented as mean ± standard error of the mean (SEM).

In Figure 3 and Figure 4, the Pearson correlation coefficient was calculated to assess the associations between two groups of parameters: baseline mitochondrial/bioenergetic characteristics and cellular responses to compound or drug treatments (see example in the case of 2-DG treatment in Appendix A). Baseline parameters included mitochondrial characteristics (such as biomass or density of mitochondrial proteins) and bioenergetic characteristics (including mitochondrial respiration, glycolysis, and bioenergetic fitness). The response parameters to treatments encompassed changes in cell number (primarily reductions post-treatment), changes in proliferation rate (mainly inhibition), and changes in apoptosis levels (primarily induction). Evaluating these correlation coefficients helps to understand the relationships, dependencies, and predictive potential between baseline cellular characteristics and drug responses.

Calculation of Weighted Combined Drug Response Correlation Score (Figure 5): To better understand the relationship between mitochondrial/bioenergetic characteristics and cell responses to drug treatments, we introduced a “combined drug response correlation score”. This score integrates the treatment-induced changes in cell number, proliferation rate, and apoptosis levels, with weighted values to reflect their significance. Specifically, the change in cell number is given a weight of 1.0, acknowledging its primary importance, while changes in proliferation and apoptosis are each weighted at 0.5. Thus, the score is calculated as the sum of these weighted correlation coefficients, represented as 1.0 × *P_CCvsBM_* (treatment-induced change in cell number vs. basal metric) + 0.5 × *P_PCvsBM_* (treatment-induced change in proliferation rate vs. basal metric) + 0.5 × *P_ACvsBM_* (treatment-induced change in apoptosis levels vs. basal metric), where *P* denotes Pearson correlation coefficients. This method provides a more precise assessment of drug effects, ranging from −2 to 2, thereby offering a broader range than the conventional Pearson coefficient scale of −1 to 1.

## Figures and Tables

**Figure 1 ijms-25-07038-f001:**
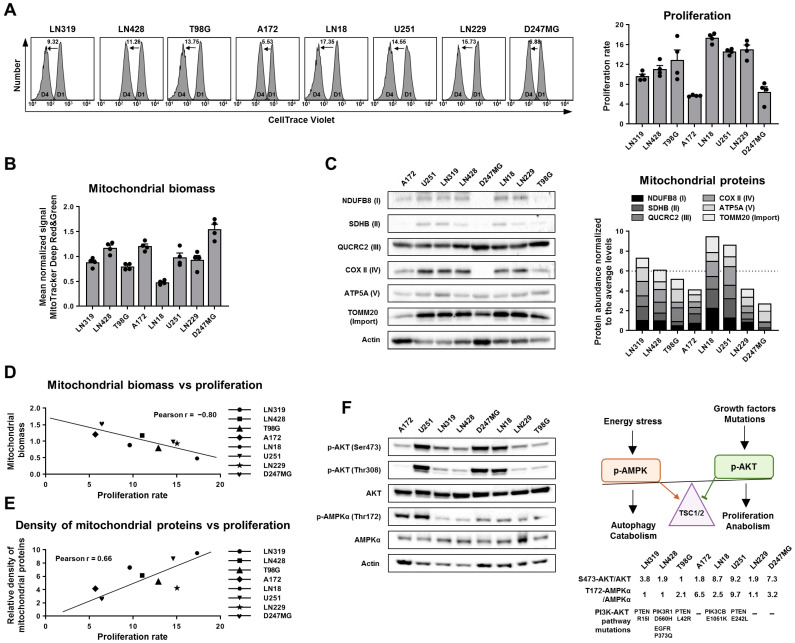
Mitochondrial biomass and density of mitochondrial proteins are linked with the proliferation rate of glioma cells. (**A**) Representative histograms and quantification of flow cytometry analysis characterizing the proliferation rate of glioma cells. The cells were prestained with 3 μM of CellTrace Violet dye and analyzed after 24 h (D1) and 96 h (D4), proliferation rate was calculated as the D1/D4 ratio. Values are means ± SEM, *n* = 3 (biological replicates). (**B**) Mitochondrial biomass quantified by flow cytometry analysis of glioma cells stained with MitoTracker Deep Red FM and MitoTracker Green FM dyes. Each dot on the graph represents the mean normalized signal from three biological replicates, each performed using one dye at a single concentration. (**C**) Western blot analysis of six mitochondrial proteins and actin in lysates of glioma cells and quantification of protein abundance represented as stacked bar charts illustrating their combined levels. Values are means, *n* = 3 (biological replicates). Error bars corresponding to three replicates and characterizing the distribution of each protein separately are presented in Appendix A. (**D**) Pearson correlation reflecting the association between mitochondrial biomass (average of mean normalized MitoTracker Green and Deep Red signals) and proliferation rate in glioma cells. (**E**) Pearson correlation reflecting the association between density of six mitochondrial proteins (Western-blot-derived mean normalized values) and proliferation rate in glioma cells. (**F**) Left part—Western blot analysis of AKT/AMPK activation in lysates of glioma cells. Right part—quantification of signal normalized to total protein.

**Figure 2 ijms-25-07038-f002:**
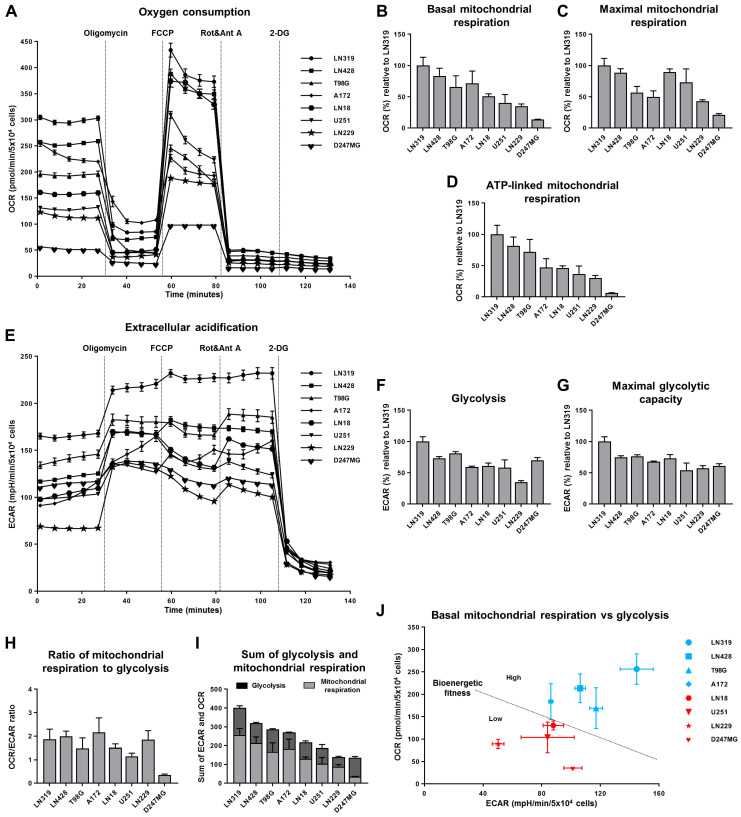
Glioma cell lines display major variability in OXPHOS levels. (**A**–**J**) Bioenergetic analysis of glioma cells with the usage of Seahorse analyzer upon subsequent injections of oligomycin, FCCP, rotenone+antimycin A, and 2-DG. (**A**–**D**) Parameters related to oxygen consumption represented by OCR values. (**E**–**G**) Parameters related to extracellular acidification represented by ECAR values. (**H**) OCR/ECAR ratio representing the relation between basal mitochondrial respiration and glycolysis. (**I**) Sums of OCR and ECAR values corresponding to basal mitochondrial respiration and glycolysis, respectively. (**J**) Visualization of OCR and ECAR values corresponding to basal mitochondrial respiration and glycolysis, respectively. (**A**,**E**) Representative graphs derived from one experiment. Values are means ± SEM, *n* = 7–9 (technical replicates). (**B**–**D**, **F**–**J**) Quantifications of 4 independent experiments (similar to (**A**,**E**)). Values are means ± SEM, *n* = 4 (biological replicates).

**Figure 3 ijms-25-07038-f003:**
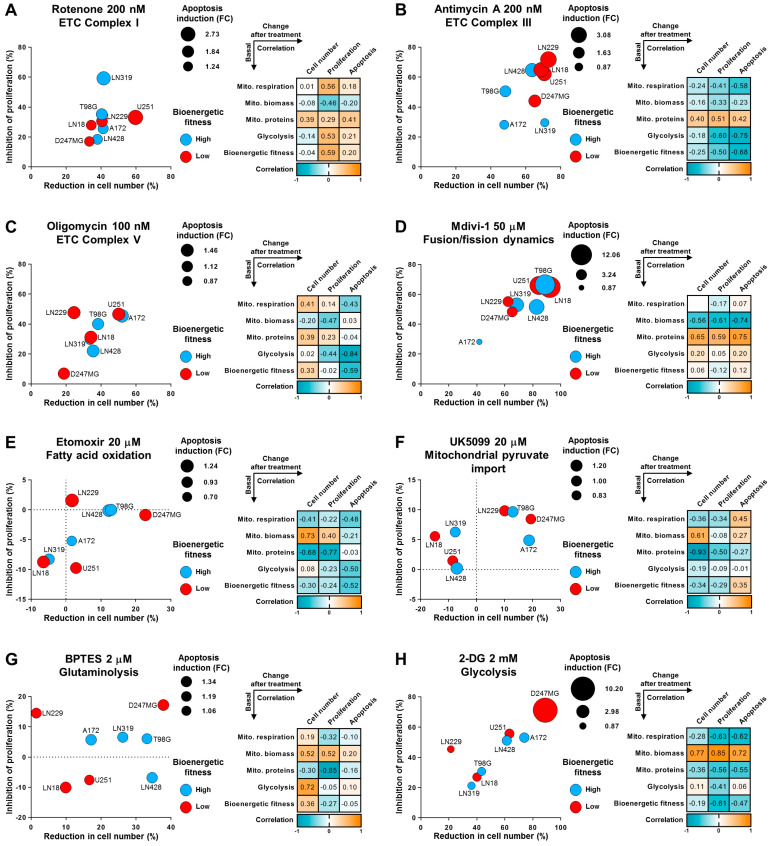
Susceptibility of glioma cell lines to mitochondria-targeting is associated with bioenergetics and mitochondrial parameters. (**A**–**H**) Glioma cells were seeded one day prior to the beginning of the experiment and then treated with selected drugs in indicated concentrations for 72 h. Subsequently, analysis of cell number, apoptosis levels, and proliferation rate was carried out (Appendix A). The obtained data were quantified and represented as fold changes or percentages over the control condition. Values are means, *n* = 3–4 (biological replicates). (**A**–**H**) **Left side:** Bubble plots illustrate the mean changes in cell number (*x*-axis, %), mean changes in proliferation rate (*y*-axis, %), and mean changes in apoptosis (size, fold change). The color coding indicates whether a cell line displayed increased bioenergetic activity (blue) or decreased (red) relative to the average level. The changes in cell number, proliferation rate, and apoptosis level were quantified using flow cytometry along with annexin V or CellTrace Violet assays. (**A**–**H**) **Right side:** Correlation matrices depict the relationships between various mitochondrial and bioenergetic characteristics measured at the steady level and the drug responses induced by treatments with selected compounds. The drug response is represented by changes in the three parameters of cell number, proliferation rate, and apoptosis.

**Figure 4 ijms-25-07038-f004:**
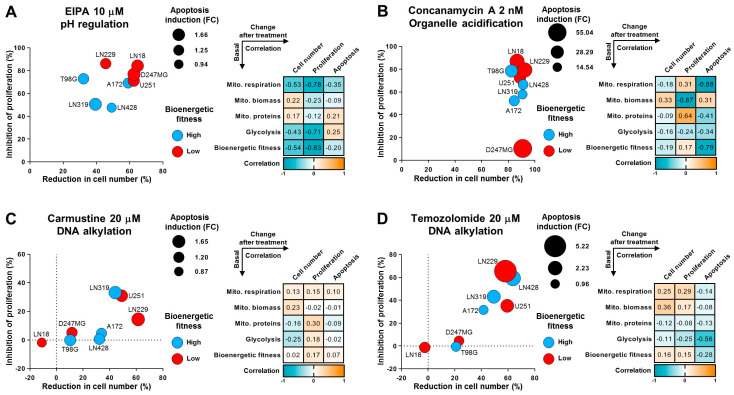
Glioma cells are sensitive to impairment of acidification regulation and the efficacy of chemotherapy treatments is associated with the density of mitochondrial proteins. (**A**–**D**) Glioma cells were seeded one day prior to the beginning of the experiment and then treated with selected drugs in indicated concentrations for 72 h. Subsequently, analysis of cell number, apoptosis levels, and proliferation rate was carried out (Appendix A). The obtained data were quantified and represented as fold changes or percentages over the control condition. Values are means, *n* = 3–4 (biological replicates). (**A**–**D**) **Left side:** Bubble plots illustrate the mean changes in cell number (*x*-axis, %), mean changes in proliferation rate (*y*-axis, %), and mean changes in apoptosis (size, fold change). The color coding indicates whether a cell line displayed increased bioenergetic activity (blue) or decreased (red) relative to the average level. The changes in cell number, proliferation rate, and apoptosis level were quantified using flow cytometry along with annexin V or CellTrace Violet assays. (**A**–**D**) **Right side:** Correlation matrices depict the relationships between various mitochondrial and bioenergetic characteristics measured at the steady level and the drug responses induced by treatments with selected compounds. The drug response is represented by changes in the three parameters of cell number, proliferation rate, and apoptosis.

**Figure 5 ijms-25-07038-f005:**
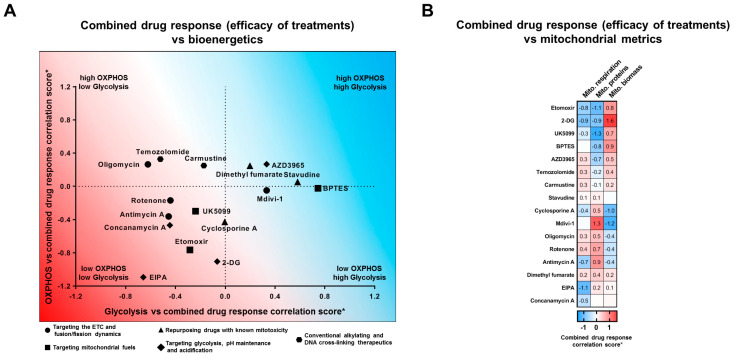
Mitochondrial parameters and bioenergetics are associated with response of glioma cells to targeting of metabolic and homeostatic pathways. (**A**,**B**) The graph and a heatmap illustrating the relationships between the bioenergetic profiles and mitochondrial characteristics of glioma cells and the efficacy of selected treatments to induce drug responses (changes in cell number, proliferation, and apoptosis). A positive score indicates that the drug response increases in line with a bioenergetic or mitochondrial characteristic (analogous to a positive correlation). Conversely, a negative score indicates that the drug response decreases when a bioenergetic or mitochondrial characteristic is high (analogous to a negative correlation). The data are derived from correlation matrices shown in corresponding figures. The treatments in the upper left part of the graph display increased efficacy against cells relying on mitochondrial respiration. In contrast, treatments in the lower right part display increased efficacy against glycolytic cells. * See Section 4 for calculation of the correlation score. All data points are derived from 3–4 independent experiments (biological replicates).

**Table 1 ijms-25-07038-t001:** Characteristics and features of eight glioma cell lines.

Name	TP53	PTEN	NF1	EGFR	IDH	PIK3R1	Sex	Origin
LN319	p.R175H	p.R15I	p.K92M, p.K92N	wt	wt	wt	M	Astrocytoma
LN428	p.R282W, p.V173M	wt	p.L1941T	p.P373Q	wt	p.D560H	M	Glioblastoma
T98G	p.M237I	p.L42R	wt	wt	wt	wt	M	Glioblastoma
A172	wt	wt	wt	wt	wt	wt	M	Glioblastoma
LN18	p.C238S	wt	wt	wt	wt	wt	M	Glioblastoma
U251	p.R273H	p.E242L	p.I679D	wt	wt	wt	M	Astrocytoma
LN229	p.P98L	wt	wt	wt	p.L13M	wt	F	Glioblastoma
D247MG	wt	wt	p.R1534T	wt	wt	wt	F	Gliosarcoma

## Data Availability

The raw data supporting the conclusions of this article will be made available by the authors on request.

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
