# Peer review of "Mitochondrial Protein Density, Biomass, and Bioenergetics as Predictors for the Efficacy of Glioma Treatments"

_ijms, 2024, doi:10.3390/ijms25137038_

Round 1

Reviewer 1 Report

Comments and Suggestions for Authors

This is a nice study on the search for new targets in glioblastoma cell lines. The authors investigate a connection between mitochondrial status/bioenergetics and the efficacy  of conventional treatments for glioma. The study is original, relevant and well defined. Is a relevant theme for health and an important problem of current interest.

Could the authors comment on the limitations of the study? i.e. i.e. human cell, but cell line (no in vivo experiments).

To complement claims in the mitochondrial role I would suggest including assays in mitochondrial structure (i.e. electron microscopy).

Reviewer 2 Report

Comments and Suggestions for Authors

The paper is interesting however, the Authors have to revise it to make it possible for publication in IJMS:

1. In figure 1 C and F we can see an uneven actin level which is concerning and it is hard to evaluate the relative expression of the other proteins. 

2. The Authors provide the plot for WB analysis without any statistical analysis and it may seem as it is done in just one replicate. Moreover, the plot is presented in an unusual way which makes it hard to understand its content. 

3. Oxygen consumption plots are also showed with nearly 0 SD's which seems as they are presenting the technical replicates. 

4. All the mitochondrial respiration data should have some control, reference set to 100% and the other values should be presented relative to control and not the raw data. 

5. Methodology for the studies on the correlation between mitochondrial proteins and mitochondria targeting capacities should be done more extensive. 

Reviewer 3 Report

Comments and Suggestions for Authors

The paper of Shaparova et al., describe the mitochondrial biomass in different glioma cell lines and correlates such peculiarity to proliferative activity and sensitivity of cells to chemotherapy drugs.

the experiments are well described and performed.

My major concerns:

1) is mitochondrial biomass only related to organelle numbers or also to michondrial dimensions?

2) how can authors normalize mitochondrial protein westerns?  It would  be more appropriate to run mitochondrial extracts and try to normalize them for a structural mitochondrial protein.

3) pAKT levels in cancer lines very often depend on mutations of PI3K pathway components.

Minor:

Authors might include a reference on the correlation of different therapeutical managements, that potentially might impact on mitochondrial pathways:

doi.org/10.1016/j.mad.2023.111801

Round 2

Reviewer 2 Report

Comments and Suggestions for Authors

The Authors made little corrections to the paper. My biggest concern is that some studies were done in a single replicate which underlines the preliminary aspect of the research... 

Author Response

Dear Reviewer,

Thank you for your attention to our manuscript.

I would like to clarify an important aspect regarding the origin of replicates. We have specified throughout the manuscript whether the data points represent technical or biological replicates.

Importantly, I want to highlight that all data shown are from biological replicates, with the exception of the panels in Figure 2A and Figure 2E. These particular panels are representative images that depict the distribution of raw data obtained from one experiment, which was repeated four times. All other graphs in this figure are derived from the quantification of these four independent biological experiments.

We appreciate your time and assistance in reviewing these details.

Reviewer 3 Report

Comments and Suggestions for Authors

The manuscript has improvedandthe results are clearly reported

Author Response

Dear Reviewer, 

We, the authors, are sincerely grateful for the insightful feedback and critique provided on our manuscript.

Thank you very much for your time and valuable assistance.

Best wishes,
The Authors